# Research of mechanical model based on characteristics of fracture mechanics of ice cutting for scientific drilling in polar region

**Xinyu Lv**[1, 3]**, Zhihao Cui**[2, 3]**, Ting Wang**[2]**, Yumin Wen**[2]**, An Liu**[4]**, Rusheng Wang**[2, 3*]

[1] Naval Architecture and Ocean Engineering College, Dalian Maritime University, Dalian 116026, China.

[2] College of Construction Engineering, Jilin University, Changchun, 130021, China

[3] Polar Research Center, Jilin University, Changchun, 130021, China

[4] Power China Huadong Engineering Corporation Limited, Hangzhou, 310014, China

**Correspondence:** Rusheng Wang (wangrs@ jlu.edu.cn)

**Abstract**:Scientific drilling in polar regions plays a crucial role in obtaining ice cores and using them to understand climate change and to study the dynamics of the polar ice sheet and its impact on global environmental changes (sea level, ocean current cycle, atmospheric circulation, etc.). Mechanical rotary cutting is a widely used drilling method that drives the cutter to rotate to cut and drill through ice layers. It is necessary to conduct in-depth research on the brittle fracture behavior of ice and mechanical model, and analyze the factors and specific mechanisms (cutter's angle, rotation speed of the drill bit, and cutting depth) affecting cutting force for the rational design of ice-core drill system, improving the efficiency of ice-core drilling, and ensuring the drilling process smoothly. Therefore, in this paper, the process of ice cutting was observed, the fracture mechanics characteristics of ice cutting process wad analyzed, the formation process of ice chips was divided into three stages, and the mathematical model for the cutting force was established based on the observation results. It describes the damage conditions of ice failure and points out the influencing factors and specific influencing laws on cutting force. Furthermore, the cutting force generated under various experimental conditions was tested. Based on typical real-time data curves of cutting force, the characteristics of cutting force were analyzed during the cutting and drilling process. Based on the comparison results of the average cutting force, the influence mechanism of various parameters on the cutting force is obtained. This proves the correctness of the mathematical model of the cutting force and provides a theoretical reference for the calculation of cutting force during ice cutting and drilling in polar regions.

## 1. Introduction

As the largest cold source on Earth, Polar ice sheets/glaciers are an important component of the
Earth's system related to the Earth's crust, glaciers, ice shelf, ocean, and atmosphere, it has a profound
impact on global changes such as climate change and sea level rise et al (Lin Yang et al., 2023). Many
scientific issues related to polar regions can be solved and validated by carrying out scientific drilling
in ice sheets and obtaining ice cores (S.H. Faria et al., 2014; P. Talalay et al., 2015; P.L. Cao et al l.,
2019). Mechanical rotary drills have been widely used in the field of polar ice core drilling (Ueda and
Garfield, 1968, 1969; Gundestrup et al., 1984; Kudryashov et al., 1994; Stanford, 1992; Wumkes, 1994;
Fujii et al., 1999; Takahashi et al.,2002; Johnsen et al., 2007; Shturmakov et al., 2007). The process of
ice core drilling mainly consists of three steps: Cutting and drilling of the ice sheet, removal and
transport of the ice chips generated at the hole bottom, and the collection of ice core and chips
precipitation (Litvinenko VS and Nikolay I Vasiliev et al., 2014). These three steps are interrelated, and
all of them have significant effects on the process of drilling. The cutting and drilling of the ice sheet
generate a cutting force, which not only affects the selection of the motor system of the drill but also
the design of the anti-torsion system, and even determines the success or failure of the cutting and
drilling of the ice sheet. By conducting in-depth research on the fracture mechanics characteristics of
solid ice, establishing a mechanical model for ice cutting, and determining the factors and specific
mechanisms affecting cutting force, it can contribute to the rational design of the drilling tool system,
the improvement of drilling efficiency, and ensure the smooth progress of drilling.
During ice core drilling, ice cutting is periodically carried out. At first, the moving cutters cut into
the ice and compress it. When the level of stress near the edge of the cutter exceeds the cutting point, a
crack is formed in the direction from the edge to the surface. This means that the horizontal force of
cutting, called $P_x$, creates a repeated series of breaks, and its value is considered to be the mean force
over the cutting length. Griffith (1920) assumed that when the energy of elastic strain exceeds the
surface energy, the existing micro-crack starts to extend like an avalanche, and the materials break.
Mellor and Sellman (1976) suggested that cutting force $P_x$ can be calculated by using specific energy
$E_S$ (N/m$^2$), which is the energy consumed per unit of cutting volume:
$$P_x = bhE_S \tag{1}$$
where b is the width of the cutter; h is the depth of cut.
Using the formula (1) to calculate cutting force is difficult because specific energy is a vague
concept. The formula ignores the influence of the structure of the cutter on the cutting force and lacks a
certain degree of practicality. Due to the difficulty of conducting strict theoretical methods for the
design of rock-cutting machines, many of the same experimental methods were developed by Mellor
(1981). Maeno (1988) assumed that in any deformation process caused by compression, tension,
bending, or cutting, the mechanics of ice failure are determined by the processes of inter/intragrain
sliding. Taking the ideal monocrystal of ice, the theoretical stress needed for the formation of sliding
zones is near 100 MPa, but for real ice, it does not exceed 0.1– 0.5 MPa (Lavrov, 1969). The
contradiction is explained by the disposition theory. According to this theory, the deformation of the
ice is determined by the defects which already exist in the ice crystal. The internal defects gradually
expand under the action of external forces, the ice destruction occurs.
Research about the calculation of cutting resistance of soil and sand is abundant, however, most of
which are empirical formulas based on an experimental basis (Jiang Pengnian, 1982). Due to the
non-uniformity, hard brittle, and the factors that affect cutting resistance are complex, most studies on
solid ice are conducted to investigate the influence of a certain variable on cutting resistance (Chiaia,
2008; S. Hell et al., 2014; A. Chao Correas et al., 2022). The in-depth study of the cutting properties of
solid ice was rarely reported.
In this paper, images of the cutting and drilling process of the ice under various experimental
conditions were captured, the fracture mechanics characteristics of the ice cutting process were
analyzed, and the formation process of ice chips was clear and divided into three stages. Based on the
result, a mechanics and mathematical model of ice cutting was built, and the influencing factors and
specific influencing laws on cutting force were analyzed. Finally, the influencing factors and laws were
verified through experimental tests. Which provides a theoretical reference for the calculation of
cutting force during ice cutting and drilling.
**2. Observation of ice fissure propagation in the process of ice cutting for ice-core drilling**
**2.1. Test stand design for study on ice cutting process**
To observe the cutting and drilling process of the ice under various experimental conditions, an ice
cutting and drilling simulation test stand has been designed (Fig. 1).

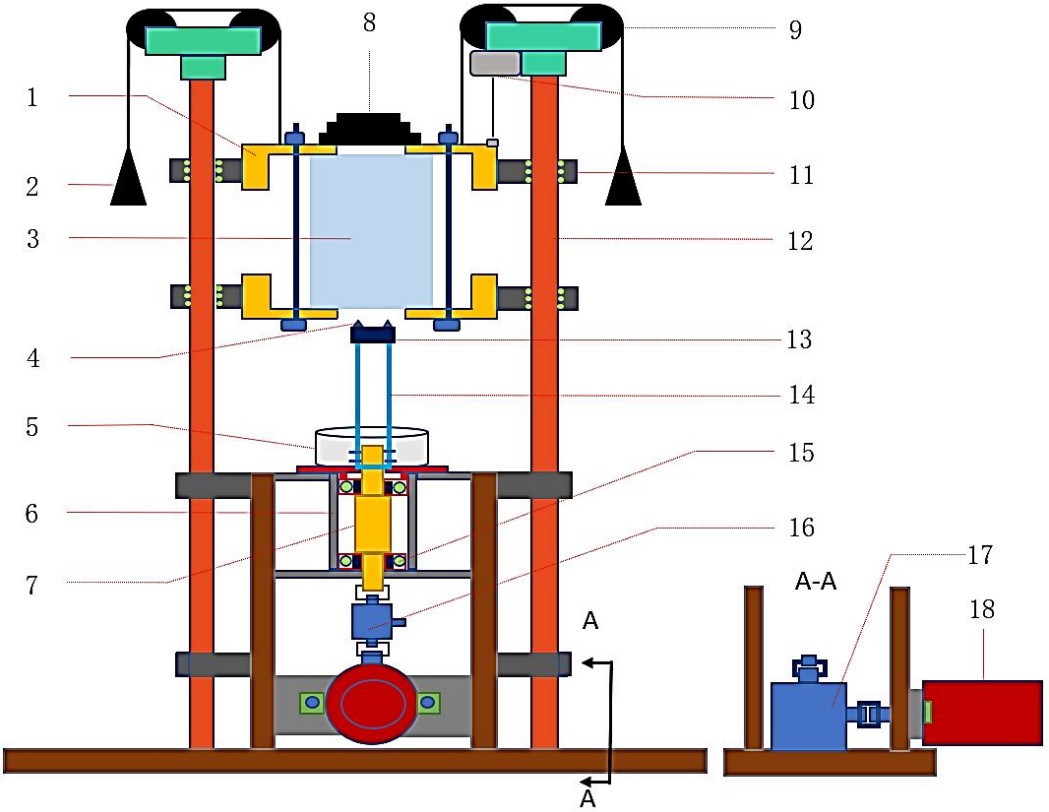


**Figure 1.** Schematic diagram of the experimental platform: 1-ice box; 2-balance weight 1; 3-ice block; 4-cutter;

5-ice chips collector; 6-cup set; 7-stepped shaft; 8-dead weight; 9-fixed pulley; 10-draw-wire displacement sensor;

11-slider; 12- slide rail; 13-drill bit; 14- drill pipe; 15- bearing; 16-torque sensor; 17- directional converter;

18-servo motor

To ensure the ice cutting and drilling proceed smoothly and the WOB is constant during the drilling

process, the drilling direction is inverted upward. Therefore, the ice chips generated in the drilling

process directly fall into the ice chips collector due to gravity, there will be no adhesion or blockage on

the drill bit. During the experimental process, the ice block and ice box can slide nearly frictionless as

they are connected to two parallel slide rails through four sliders, and the slider is equipped with rolling

balls inside to ensure that the slider slides almost frictionless on the slide rail. So, during the drilling

process, constant drilling pressure can be ensured, and multiple drilling pressure tests can be achieved

by increasing or decreasing balance weight and dead weight. The drill pipe, drill bit, and cutters are

driven to rotate by the servo motor system, and its rotation speed can be adjusted arbitrarily between

0-1000rpm. In this way, the adjustment of the rotation speed of the drill bit is achieved. The cutter

equipped in the experimental test stand can be replaced arbitrarily according to the experimental

requirements, therefore, it is possible to conduct cutting and drilling tests on cutters with various
structures.
During the experiment, the torque generated by driving the rotation of drill pipes, step shafts, and
other components, as well as the cutting torque generated by ice cutting is measured by the torque
sensor. Before conducting the cutting and drilling experiment, adjust the rotation speed of the drill bit
to the rotation speed for the next experiment, and let the drill bit and other components blank run. After
the torque measured by the torque sensor stabilizes, the torque is recorded as $T_1$. Next, perform cutting
and drilling. After the cutting and drilling process stabilizes, the recording of cutting torque begins.
after the drilling process, the average cutting torque during this period is recorded as $T_2$. So, the torque
for ice blocks cutting $T_c$ can be calculated according to the following formula (2).
$$T_c = T_2 - T_1 \qquad\qquad (2)$$
The drilling depth and time are measured by the Draw-wire displacement sensor. The formation
process of ice chips is captured by a high-speed camera.
**2.2. Test stand building and observation of ice fissure propagation during ice-core drilling testing**
Based on the above working principle, the ice core drilling test stand has been established (Fig. 2)**.**

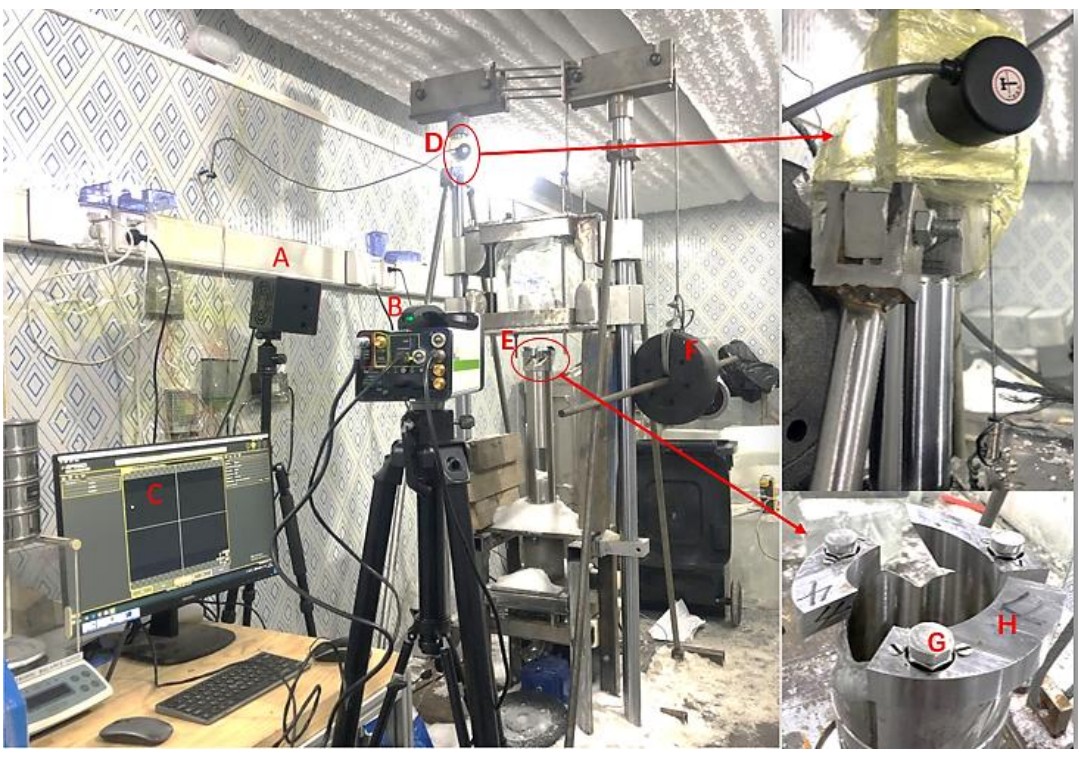


**Figure 2.** Test stand: A-light source; B-high speed camera; C-image display computer; D-draw-wire
displacement sensor; E-drill bit; F-counterweight block; G-drill bit shoe; H-cutter
The specific parameters of the main equipment in the test stand are shown in Table 1.
**Table 1.** Main parameters of equipment

| equipment and sensor | | Model | Main parameters |
|---|---|---|---|
| Servo motor system | Driver | 3DM2080-DSP | Drive voltage: 130-220VAC |
| | | | Pulse mode: Mono pulse |
| | | | Adjustment range: 0-1000rpm |
| | Motor | 130BYG350D | Maximum output torque: 60N.M |
| | | | Step angle: 1.2° |
| | | | Rated voltage and current: 220V and 8.5A |
| | Pulse generator | CS10-3 | Output mode:Steering + pulse |
| | | | Adjustment range: 0-1000rpm |
| | | | Output signal voltage: 5V; Power range:9-30V |
| Torque sensor | | LLBLS-I | Measuring range: 60N.M: Overall accuracy:0.3% |
| | | | Maximum speed: 6000rpm |
| Draw-wire displacement sensor | | MPS-M | Measuring distance:0-1500mm |
| | | | Resolving power:0.01mm; |
| | | | Pulling force of stay wire:4N |
| Slide rail and slider system | | Ø50; SK50 | Friction coefficient: 0.0010-0.0015 |
| High-speed camera | | Ispeed-7 | Technology: CMOS active pixel |
| | | | Resolution: 2048×1536 |
| | | | Frames per second: 1000000fps |
| | | | Shutter: 1us |
| | | | Lens options: F mount/G mount/C mount |

Before the experiment, the cutters (Fig.2.H) made from tool steel (W18Cr4V) shall be installed on
the drill bit (Fig.2.E) through bolts and pins (Fig.2.H) that also serve as the shoes with adjustable height.
The height of the bolts is lower than the height of the cutter's tip when the ice block slides into contact
with the shoes, the cutters have been cut into the ice block at the designed depth. Thus, the cutting and
drilling at the designed cutting depth is realized and the cutting depth has been accurately controlled.
Aiming the high-speed camera (Fig.2.B) at the cutting edge of the cutter, adjusting the frame number
of the high-speed camera to 100,000, meanwhile, supplementing the light on the object with the light
source (Fig.2.A), until the image displayed in the computer (Fig.2.C) is clear. After the experiment, the
images of the formation process of ice chips are captured and saved in a high-speed camera. The
observation experiment of the cutting and drilling process is conducted under various experimental
conditions (multiple cutter angles, cutting depths, and rotation speed of drill bit). The specific
parameters of experimental conditions are shown in Table 2. The cutter used in the experimental
process are processed with wire cut technology. Before the experimental, to prevent the impact of
surface burrs, slag, and surface roughness on the test results, sandpapers with gradually decreasing
particle size ware selected to manually polish the surface of the cutter until it was smooth. After each
test, the surface and cutting edge of the cutter are observed, if there is wear or damage, the cutter is
polished or replaced directly. The cutters- tested in the experiment are shown in Fig. 3.
**Table 2.** The specific parameters of experimental conditions

| Structure of cutter | | | Cutting depth (mm) | Rotation speed (rpm) | Ice sample dimension (mm) | Ice core diameter (mm) |
|---|---|---|---|---|---|---|
| Width (mm) | Rake angle (°) | Relief angle (°) | | | | |
| 25 | 20 | 5 | 1 | 50 | ~250×250×450 | 60 |
| | 30 | 10 | 2 | 100 | | |
| | 40 | 15 | 3 | 150 | | |

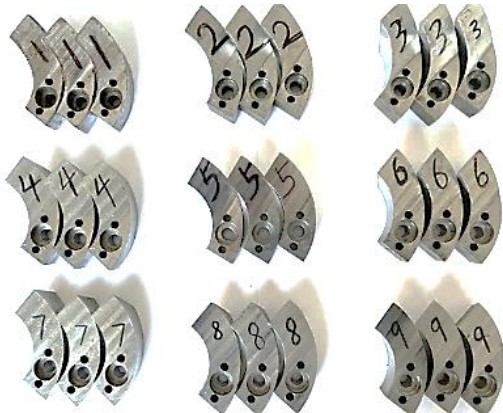


**Figure 3.** Multi-group structure cutters
This study mainly focuses on the establishment of mechanical model during the ice cutting and
drilling process. And, studies have shown that the crystal orientation, the crystal size and the density of
ice samples in NGRIP boreholes in Greenland are similar to naturally formed and artificially frozen ice
samples (Center for Ice and Climate, 2023; Cuffey and Paterson, 2010). Moreover, many scholars have
conducted experiments on artificially prepared or naturally formed ice samples, and have ultimately
obtained convincing experimental data and conclusions, providing valuable references for research in
the polar field (Narita et al., 1994; Talalay, 2003; Hong et al.,2015; Wang et al, 2024). In order to
better observe the formation process of ice chip, at present stage, this study selected transparent ice and
explored the fracture process and cutting force generated by this type of ice. The ice with variety
properties belongs to brittle materials, and there will be similarities in the fracture process. In the future,
the cutting and drilling experiments with different ice sample properties to explores the effect of ice
properties against the cutting force will be carried out. The ice blocks used in this experiment are
frozen by an ice-making machine (Fig.4), which can produce transparent ice samples without bubbles.
Then, we divided these blocks into experimental ice blocks with uniform dimensions (Fig.5) of
$\sim 250 \times 250 \times 450$ mm. and all tests were carried out in the refrigerated container with a constant
temperature of -15℃.

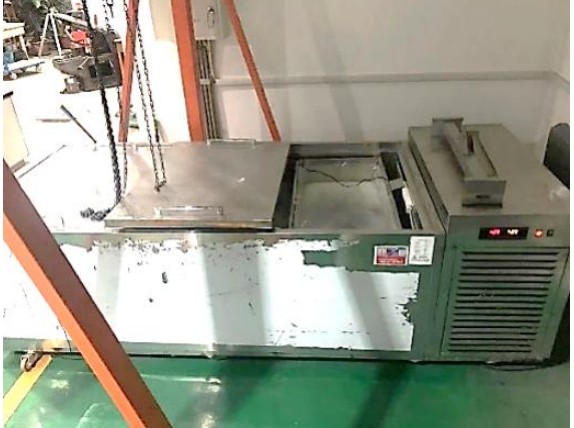 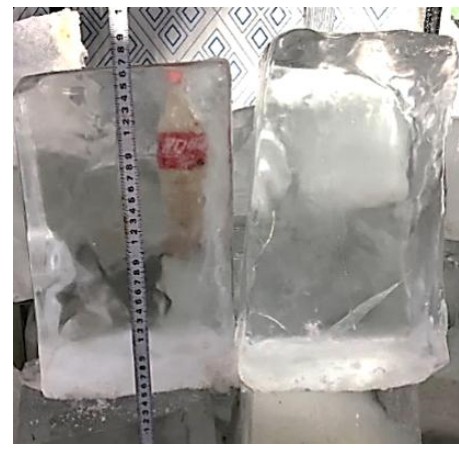

**Figure 4.** Ice-making machine                 **Figure 5.** Experimental ice samples
**3. Analysis of characteristics of ice fracture mechanics in the process of ice cutting**

It is preliminary observed after the mechanical testing of ice under the special experimental

condition. The actual ice-cutting process captured by a high-speed camera is shown in Fig. 6.

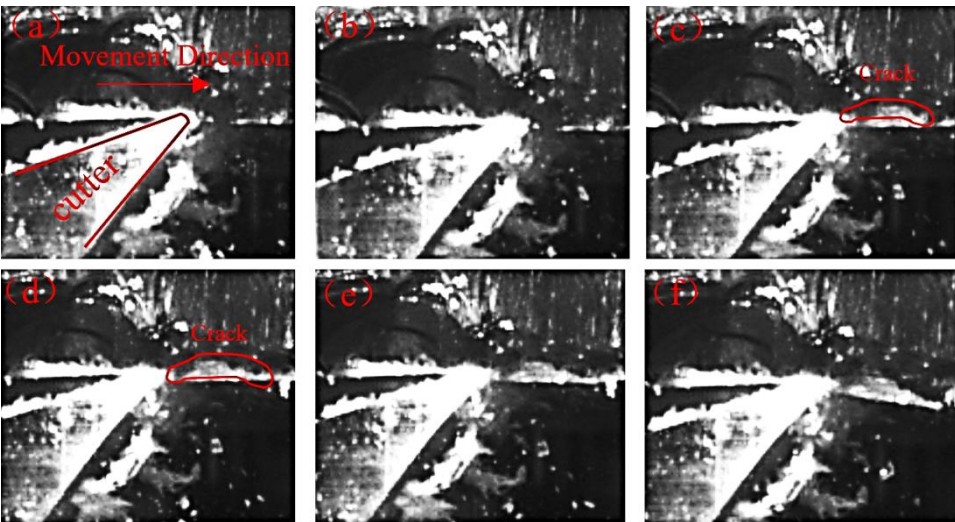


A (rake angle is 40°, relief angle is 15°, cutting depth is 1 mm and the rotation speed of the drill bit is

100 rpm)

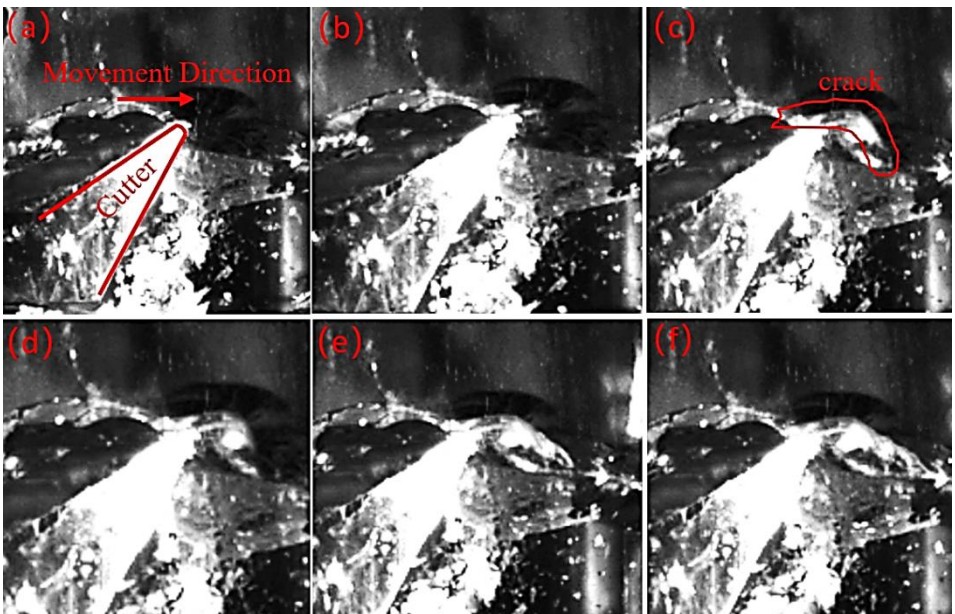

B (rake angle is 30°, relief angle is 25°, cutting depth is 2 mm and the rotation speed of the drill bit is

100 rpm)

**Figure 6.** Cutting process captured by high-speed video camera
Compared with the cutting and drilling process at a cutting depth of 1mm, when cutting and drilling
at a cutting depth of 2mm, the depth of the cutter inserted into the ice sample increases resulting in
more small particle ice chips.The particle size of the ice chips formed by major fracture increases, and
the surface after cutting becomes more uneven.
Under various experimental conditions, the ice cutting process is similar. In the cutting process, the
cutting of the ice is constantly repeated, the main damaged form of ice is a brittle fracture, the chips
show wedge block with no significantly deform, and wedge-shaped ice chips with different particle
sizes are constantly formed under variety experimental conditions. The formation process of a single
large particle of ice chips can be divided into three stages. In the first stage, the cutter invades the ice,
and the ice is compressed by the rake and relief surfaces of the cutter, resulting in ice crushing and
smaller ice chip formation (Fig. 6. a, b). In the second stage, with the rotation of the drill bit, cracks
appeared in the ice, and the cracks began to expand along a direction that approximately presented an
angle of 45° with the horizontal direction (Fig. 6. c, d). However, there were no gaps or separations
between the ice and cutters on both sides of the cracks. In the third stage, the cutter moved forward, the
crack expanded to form ice chips with large particle size that slid forward, and finally detached from
the ice. At the same time, ice chips with small particle sizes were also generated on the sliding surface
(Fig. 6. e, f).
**4. Study on a mechanical model of ice cutting process**
**4.1. Mechanical model building based on the characteristics of ice fracture mechanics**
According to the observation results of the ice cutting, it can be considered that the damage of the ice
is the result of shear slip failure caused by the compression effect of the cutter. In this process, the force
exerted on the ice chips mainly includes the squeezing force $F_n$ towards the ice, and along the normal
direction of the cutter's rake face. The frictional force $F_m$ exerted by the cutter when the ice chips
flow out; At the same time, the shear surface of the ice will also be subjected to normal pressure $F_{ns}$
and shear force $F_s$. Before the cutting of the ice, these two pairs of forces are in equilibrium. The
relationships between these forces are analyzed in front of the cutting edge (Fig. 7).

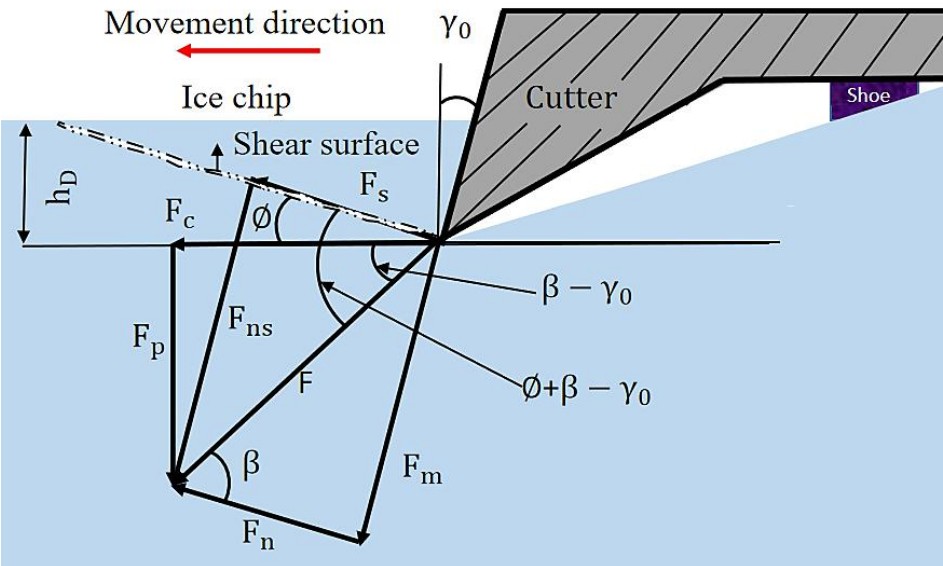


**Figure 7.** Relationship between force and angle
Where F is the combined force of $F_m$ and $F_n$, Ø is the shear angle, β (Friction angle) is the angle
between $F_n$ and F, $\gamma_0$ is the rake angle of the cutter, $F_p$ is the component force perpendicular to the
movement direction of the cutter, which is applied to the cutter and mainly provided by the weight on
drill bit during the ice layer cutting and drilling, causing the cutter to cut into the ice to a certain depth.
During the cutting and drilling process, the cutter comes into contact with the ice sample before the
shoes. Only when the cutter is inserted into the ice layer with designed cutting depth, the shoes will
fully contact the bottom of the borehole. Prior to this, there will be continuous $F_p$ on the cutter. As the
drill bit rotates, the cutter always inserts the ice sample before the shoe, and the $F_p$ on the cutter will
continue to exist. Where $F_c$ is the component force acting on the ice layer, and during the ice layer
cutting and drilling process, this force is mainly provided by the motor, which is called the cutting
force. $h_D$ is the cutting thickness. If the cutting width is represented by $b_D$, The cutting width
represents the width of the annular gap between the ice core and the hole wall in the process of ice
drilling (cutting width, width of the cutter), The area of the nominal cross-section of the cutting layer is
represented by $A_D$ $(A_D = h_D b_D)$, The area of shear surface is represented by $A_s$ $(A_s = A_D/sin\emptyset)$, the
shear stress on the shear plane is represented by τ, then
$$F_s = \tau A_s = \frac{\tau A_D}{sin\emptyset} \qquad (3)$$

According to Fig. 7, it can be concluded that:

$$F_s = F\cos\,(\emptyset + \beta - \gamma_0) \qquad (4)$$

According to the relationship between various forces, it can be concluded that:
$$F = \frac{F_s}{\cos\,(\emptyset+\beta-\gamma_0)} = \frac{\tau A_D}{sin\emptyset\cos\,(\emptyset+\beta-\gamma_0)} \qquad (5)$$

$$F_p = F\sin\,(\beta - \gamma_0) = \frac{\tau A_D \sin\,(\beta-\gamma_0)}{sin\emptyset\cos\,(\emptyset+\beta-\gamma_0)} \qquad (6)$$

$$F_c = F\cos\,(\beta - \gamma_0) = \frac{\tau A_D \cos\,(\beta-\gamma_0)}{sin\emptyset\cos\,(\emptyset+\beta-\gamma_0)} \qquad (7)$$

**4.2. Analysis of factors influencing cutting forces via the mechanical model**
According to Fig. 7, there is no shear stress in the plane perpendicular to the combined force F, so
the main stress is completely determined by the F. The material is in the state of plane stress, and the
included angle between the direction of the maximum shear stress and the direction of the maximum
principal stress is 45°, the included angle between the maximum principal stress and the F is 45 °, then
there is:
$$\emptyset + \beta - \gamma_0 = \frac{\pi}{4} \qquad (8)$$

So:
$$\emptyset = \frac{\pi}{4} - \beta + \gamma_0 \qquad (9)$$

The shear angle $\emptyset$ is affected by the rake angle of the cutter $\gamma_0$ and friction angle β. As the rake
angle of the cutter $\gamma_0$ increases, the shear angle $\emptyset$ increases; as the friction angle β increases, $\emptyset$
decreases.
The area of the nominal cross-section of the cutting layer is represented by $A_D$ ($A_D = h_D b_D$), The
area of the shear surface is represented by $A_s$ ($A_s = A_D/sin\emptyset$), the shear stress on the shear plane is
represented by $\tau$, then, according to equation (5) and the relationship between the nominal
cross-section and the shear plane, it can be obtained that:
$$F_c = \frac{\tau A_D \cos\ (\beta - \gamma_0)}{sin\emptyset \cos\ (\emptyset + \beta - \gamma_0)} \quad (10)$$

When the ice is about to break, the shear stress on the shear plane reaches its maximum value. This
value is determined by the properties of the ice and will not change as the drilling conditions. Therefore,
the cutting force is influenced by the cutting width of the cutter and the cutting depth. The cutting force
shows a linear increasing trend with the increase of the cutting width and the cutting depth. In addition,
the cutting force is also affected by the shear angle $\emptyset$、 friction angle $\beta$, and cutter's rake angle $\gamma_0$. The
friction angle $\beta$ is a certain value as the properties of the ice and cutter's material. The shear angle $\emptyset$ is
determined by the friction angle and the cutter's rake angle as shown in formula (9). Substituting
equation (9) into (10) and solving for the combined cutting force $F_c$, the following equation can be
given:
$$F_c = \frac{\tau h_D b_D \cos\ (\beta - \gamma_0)}{\sin\ (\frac{\pi}{4} - \beta + \gamma_0)\ \cos\ (\frac{\pi}{4})} \quad (11)$$

After simplifying the above equation, it can be obtained that:
$$F_c = \frac{2\tau h_D b_D}{1 - \tan\ (\beta - \gamma_0)} \quad (12)$$

It can be seen from the formula (12) that the factors affecting the cutting force mainly consist of four
sides: The first aspect, it related to the shear strength of the ice, with the increase of shear strength, the
cutting force increases gradually. The second aspect, it influenced by the cutting depth, with the
increase of cutting depth, the cutting force increases gradually. The third aspect, it affected by the
cutting width, with the increase of cutting width, the cutting force increases gradually. Finally, the rake
angle of the cutter also has an impact on the cutting force. Formula (12) shows that: within the $\beta - $
$\gamma_0 \le \frac{\pi}{2}$ range, as the rake angle of the cutter $\gamma_0$ increases, $\beta - \gamma_0$ gradually decreases, and the
$\tan\ (\beta - \gamma_0)$ decreases, $1 - \tan\ (\beta - \gamma_0)$ increases, $F_c$ decreases.
**5. Test on the characteristics of cutting force and its influencing factors for verifying the**
**mechanical model**
**5.1 Analysis of the characteristics of cutting force**
To verify the theoretical analysis results of the factors affecting cutting force, the cutting torque
collected by the torque sensor under various cutter angles, rotation speed of the drill bit, and cutting
depth conditions were measured.
After the experiment, the torque for ice cutting and drilling can be obtained through formula (2). The
schematic diagram of the torque and cutting force generated during the ice cutting drilling process is
shown in Figure 8, The relationship between the cutting force $F_c$ generated by cutting the area of the
circular ring and the torque $T_c$ measured by the torque sensor is as follow.
$$T_c = F_c r_A \tag{13}$$

Where $r_A$ is the average radius of the circular ring.

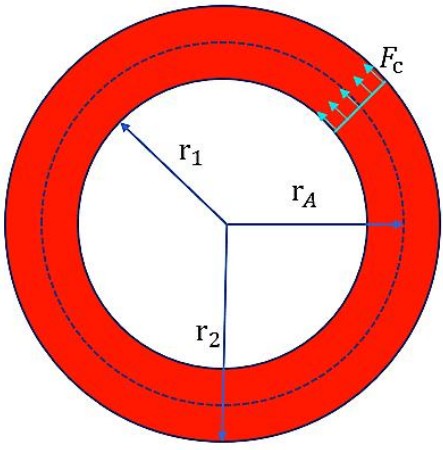


**Figure 8.** The schematic diagram of the torque and cutting force
By processing the data collected by the torque sensor, the cutting force generated by one cutter
during the ice block cutting and drilling is obtained. The typical cutting force trace generated during the
ice cutting process is shown in Fig. 9.

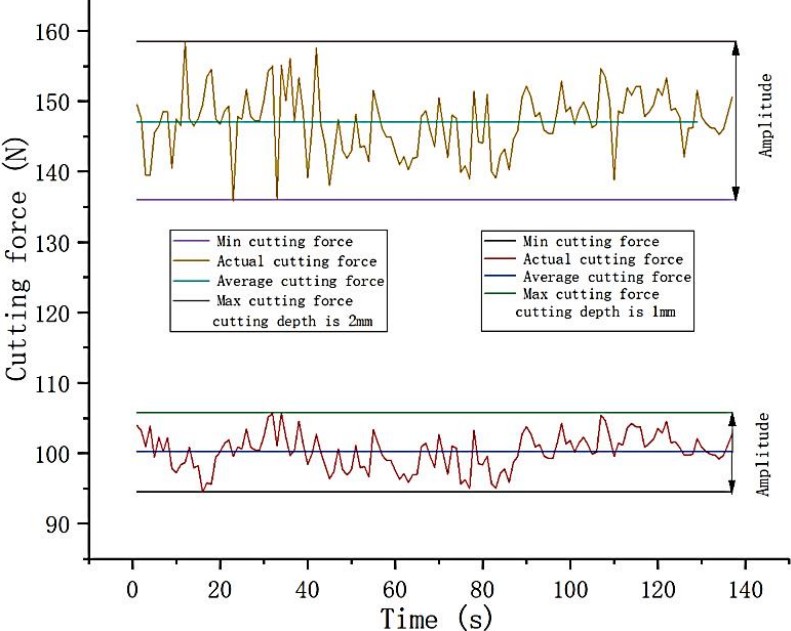


**Figure 9.** Typical cutting force trace(Cutting depth is 1 mm and 2 mm; Rotation speed of drill bit is 50rpm; Rake angle is 30°; Relief angle is 5°)

Fig. 9 shows the cutting force trace generated during two cutting and drilling process, which were carried out under the same conditions except for cutting depth, both cutting force traces oscillate at a certain frequency within a certain range, and the oscillation consists primarily of two frequencies, in addition the oscillation frequencies of two cutting force trace are similar. The higher frequency is related to the resolution of the sensor. The sensor outputs data at a certain interval during the recording process, the output data is not continuous, resulting in fluctuations in the trace. The lower frequency is related to the formation of large particle ice chips. Unlike ductile materials, where the chips produced by a shearing action are continuous and the forces appeared relatively constant, chips from brittle materials are produced by a repeated series of breaks. When the cutter is pressed into the ice, the cutting force begins to rise and elastic energy is stored in the cutter assembly, some of the energy is expended in local crushing, the ice layer undergoes shear-slip deformation. As the cutting force reaches a magnitude necessary to induce a major fracture, a crack propagates into the ice, releasing the cutter elastic energy and dislodging a major chip, the force than suddenly decreases. Therefore, during the cutting and drilling process in the ice layer, the cutting force trace exhibits an oscillating state, the amplitude of the oscillation is related to the cutting depth. During the process of the cutting depth increase, the degree of rapid increase and decrease in cutting force will be more severe. As show in Fig.

9, when drilling with a cutting depth of 2 mm, the oscillation amplitude of cutting force is greater than
that of drilling with a cutting depth of 1 mm.

As the cutting depth increases, the degree of crack propagation into the ice will also increase. When

the crack extends into the ice core, it will cause a decrease in the surface quality of the ice core. It is
necessary to control the cutting depth reasonably during the cutting process to ensure the quality of the
ice core. Moreover, the study results on mechanical models of ice cutting process indicated that:
"within the range of $\beta - \gamma_0 \leq \frac{\pi}{2}$, the cutting force gradually decreases with the increase of the rake
angle". The rake angle can be appropriately increases within this range to reduce the oscillation.
**5.2. Test of the factors influencing cutting force**

After the cutting and drilling experiments, the average cutting force was obtained under each

experimental condition. Plots of the average cutting force versus the cutter's rake angle are shown in
Fig. 10.

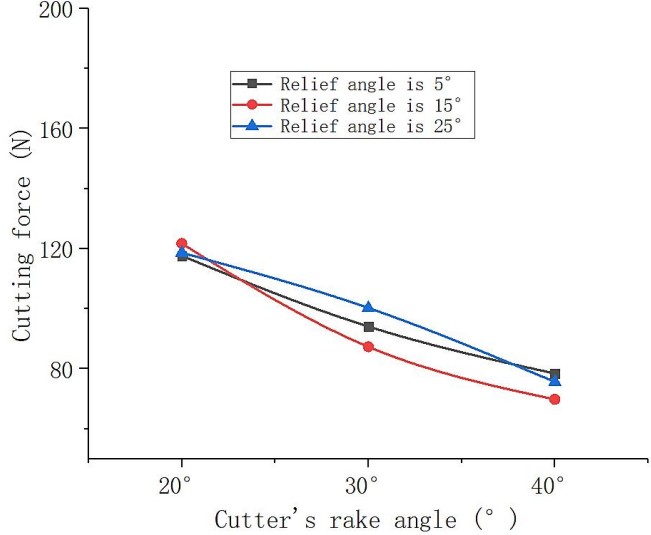


(a) The cutting depth is 1 mm, and the rotation speed of the drill bit is 100rpm

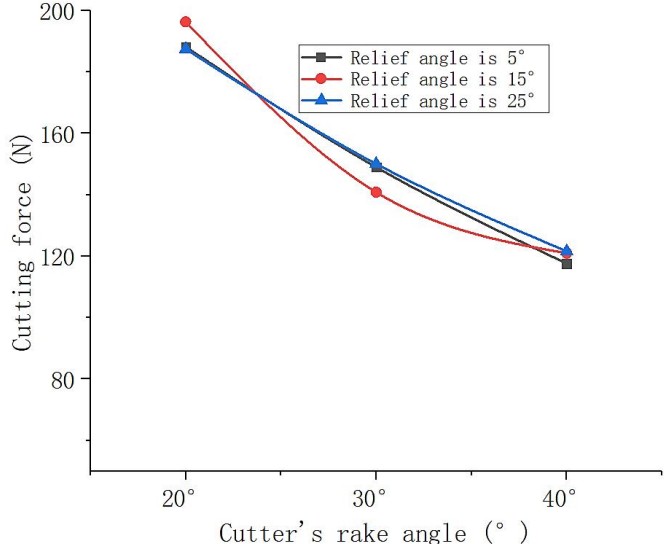


(b) The cutting depth is 2 mm, and the rotation speed of the drill bit is 100rpm

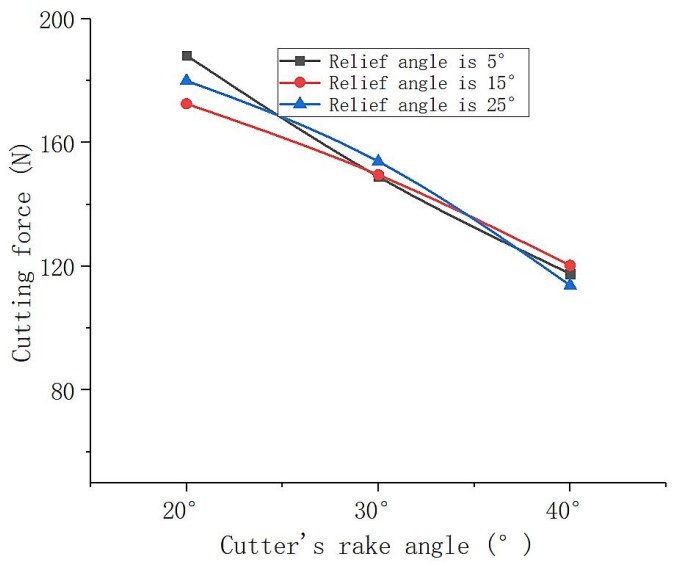


(c) The cutting depth is 2 mm, and the rotation speed of the drill bit is 50rpm

**Figure 10.** Cutting force versus cutter's rake angle

As shown in Fig. 10, when the cutting depth is 2mm, the rotation speed of the drill bit is 100 rpm,

and the rake angle of the cutter is 20 °, the cutting force reaches the maximum value of 196.3451N.
When the cutting depth is 1mm, the rotation speed of the drill bit is 100 rpm, and the rake angle of the
cutter is 40 °, the cutting force reaches the minimum value of 69.83529N. The cutting force varies
within this range under the other experimental conditions. That is, under various cutting depths and
drill speed conditions, the cutting force gradually decreases with the increase of the cutter's rake angle.

Plots of the average cutting force versus the cutter's relief angle are shown in Fig. 11.


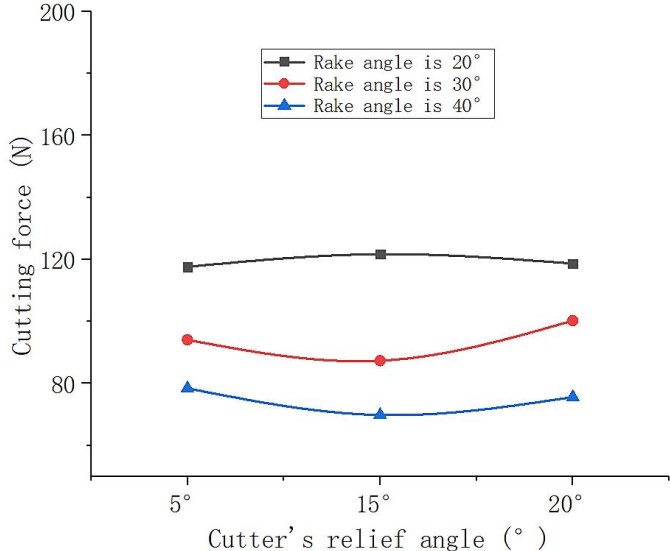

(a) The cutting depth is 1 mm, and the rotation speed of the drill bit is 100rpm

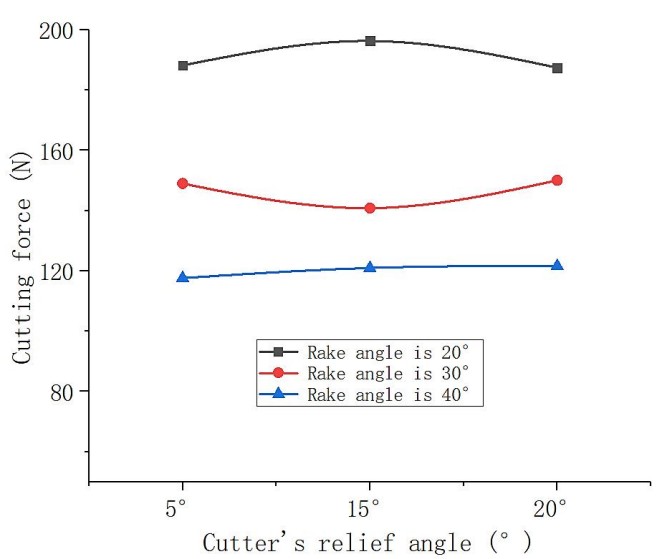


(b) The cutting depth is 2 mm, and the rotation speed of the drill bit is 100rpm

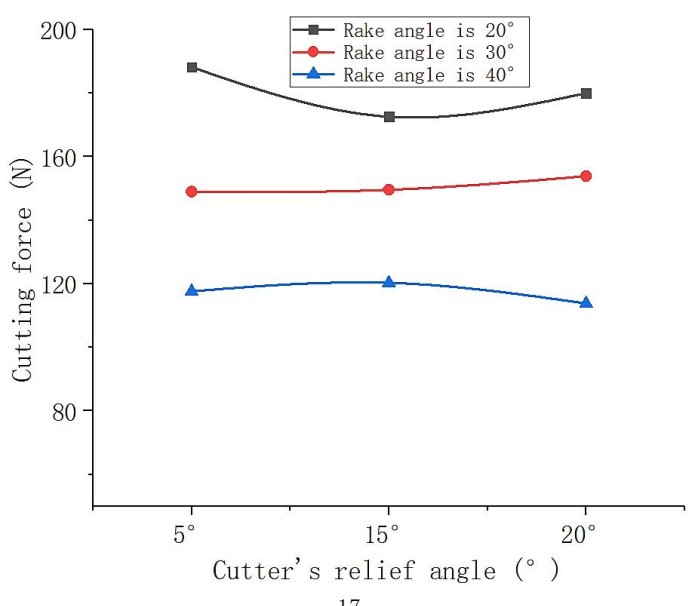


324      (c) The cutting depth is 2 mm, and the rotation speed of the drill bit is 50rpm

325  **Figure 11.** Cutting force versus cutter's relief angle

326   Under various experimental conditions, the relief angle of the cutter changes, and the cutting force

327  only changes slightly. Moreover, with the change of the relief angle of the cutter, the cutting force does

328  not show a clear and consistent change pattern. Therefore, it can be inferred that the relief angle of the

329  cutter has no clear effect on the cutting force.

330   Plots of the average cutting force versus the rotation speed of the drill bit are shown in Fig. 12.

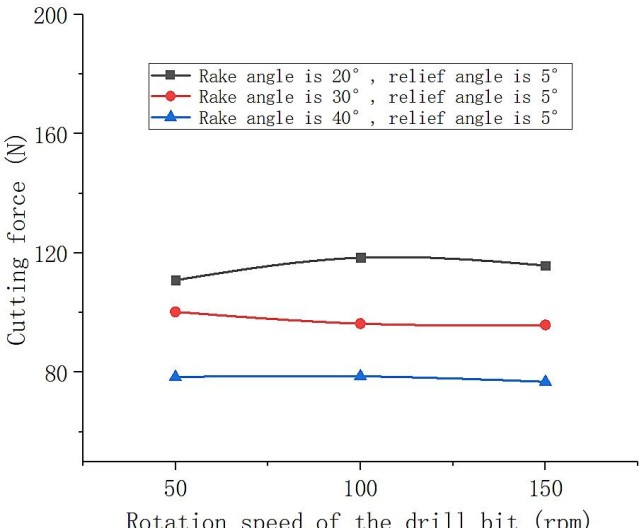


332          (a) The cutting depth is 1mm

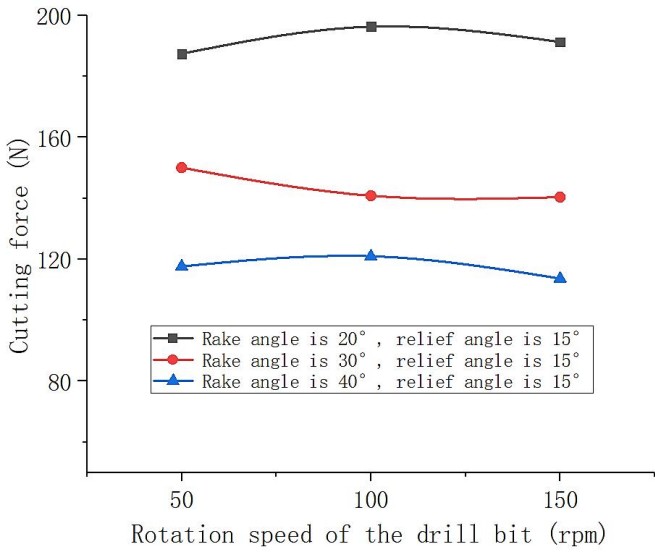


334          (b) The cutting depth is 2mm

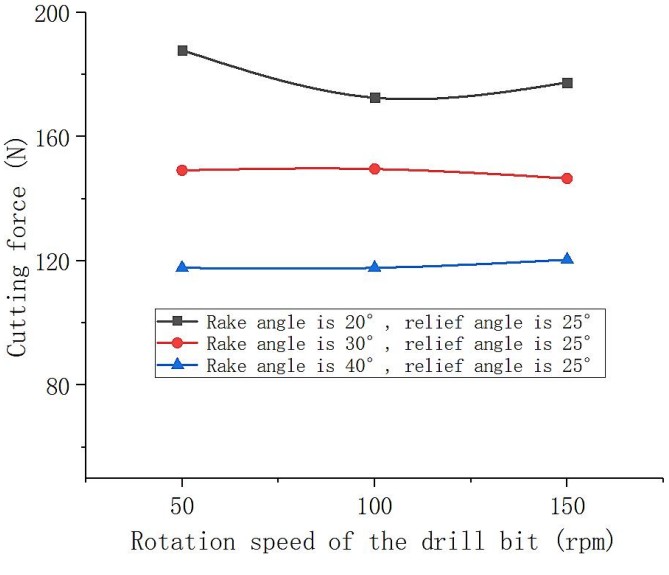


(c) The cutting depth is 2mm
**Figure 12.** Cutting force versus rotation speed of the drill bit
Under various experimental conditions, there is only a slight change in cutting force during the process of the
rotation speed changing, and there is no clear pattern of change. The rotation speed of the drill bit does not affect
the cutting force.
Plots of the average cutting force versus cutting depth are shown in Fig. 13.

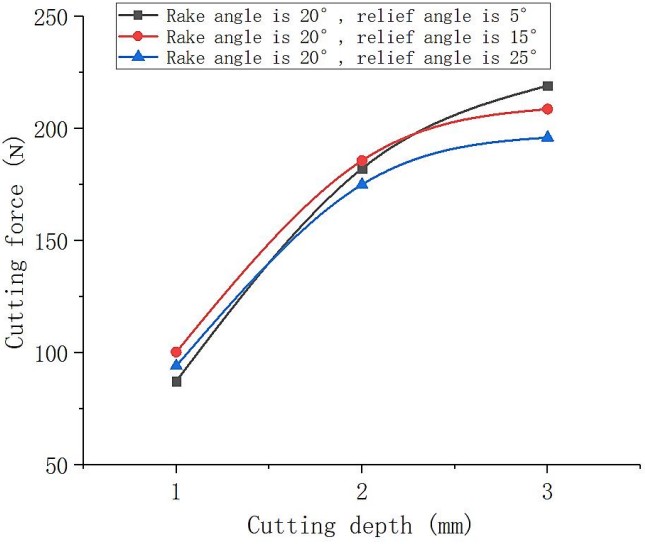


(a) The rotation speed of the drill bit is 50 rpm

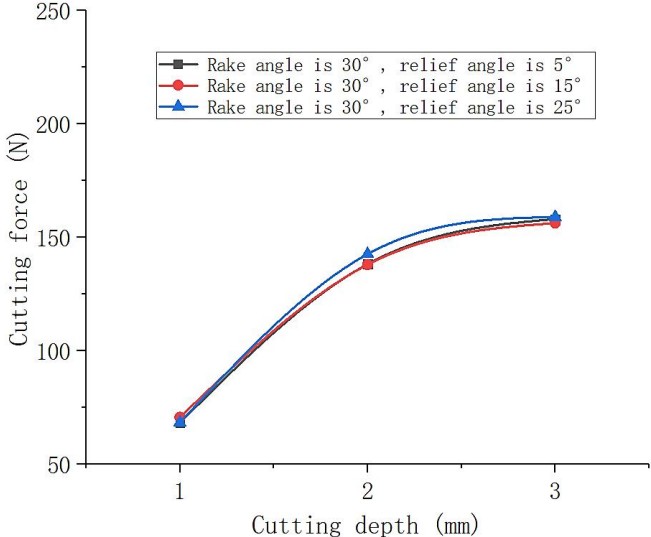


(b) The rotation speed of the drill bit is 100 rpm

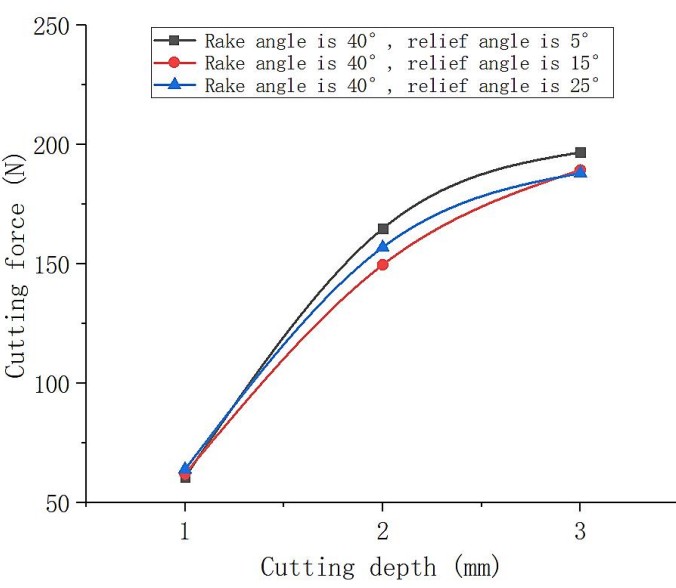


(c) The rotation speed of the drill bit is 100 rpm
**Figure 13.** Cutting force versus cutting depth
Under all experimental conditions, as the cutting depth increases, the cutting force shows a gradually
increasing trend. When the cutting depth is 3mm, the maximum cutting force reaches 219.13725N. And,
under the same experimental condition, the cutting depth increasing from 1 mm to 2 mm results in an
approximate doubling of the cutting force. As the depth of penetration increases, the cutting force
continues to increase, but the increasing trend gradually weakens.
**6. Conclusions**
It is preliminarily observed after the mechanical testing of ice, that the main damage form of ice is a
brittle fracture in the cutting process. During this process, the cutters press into the ice to a certain
depth and rotate, the ice withstands a squeezing effect from the rake face of the cutter and the shear slip
deformation occurs. When the shear slip deformation reaches a certain degree, the ice undergoes shear
failure and then forms ice chips. This process is constantly repeated throughout the cutting and drilling
of the ice.
Based on the characteristics of ice cutting and the stress characteristics during the ice cutting and
drilling process, a mechanical model of ice cutting was established. The mechanical model shows that
the cutting force is not only affected by the mechanical properties of ice but also by the cutting width,
cutting depth, and the rake angle of the cutter. As the cutting width and cutting depth increase, the
cutting force increases; as the increase of rake angle of the cutter, the cutting force decreases.
Additionally, the characteristics of cutting force were analyzed through experimental methods. The
experimental results show that the cutting force traces oscillated within a certain range, the oscillation
consists primarily of two frequencies. The higher frequency is related to the resolution of the sensor,
the lower frequency is related to the formation of large particle ice chips. the oscillation amplitude of
the cutting force traces is related to the cutting depth, as the cutting depth increases, the oscillation
amplitude of the trajectory will also increase. In addition, the oscillation amplitude will also affect the
quality of the core, as the amplitude increases, the possibility of the ice core breaking will also increase,
and the quality of the ice core will also increase accordingly. It is necessary to control the cutting depth
reasonably during the cutting process to ensure the quality of the ice core. Finally, the influencing
factors and laws of cutting force were verified by analyzing the cutting force generated under various
experimental conditions.

*Date availability*.  No data sets were used in this article

*Author contribution.* Xinyu Lv: Conceptualization, Methodology, Writing – original draft; Zhihao Cui:
Methodology, Validation, Formal analysis, Visualization; Ting Wang: Methodology, Validation,
Formal analysis, Visualization; Yumin Wen: Conceptualization, Writing – review & editing,
Methodology, Validation; An Liu: Methodology, Validation, Formal analysis, Visualization; Rusheng
Wang: Methodology, Formal analysis, Supervision, Project administration, Funding acquisition.

*Competing interests.* The contact author has declared that neither of the authors has any competing
interests.

*Disclaimer. Publisher's note:* Copernicus Publications remains neutral with regard to jurisdictional
claims in published maps and institutional affiliations.

*Special issue statement.* This article is part of the special issue "Ice core science at the three poles
(CP/TC inter-journal SI)". It is not associated with a conference.

*Financial support.* This paper presents research conducted with support from the National Key R&D
Program of China (Project No. 2021YFA0719100; Subject No. 2021YFA0719103), and Jilin
University "Interdisciplinary Integration and Innovation" project (Project No. 419021421601).

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
