# Peer review of "Research of mechanical model based on characteristics"

_EGUsphere, 2023_

## Author Comment (AC1)

Dear Referee,

Thank you for your response, and pass our great thanks to you for fruitful comments and advises. We tried to consider all mentioned comments and hereafter explain every change made point by point. The comments are in black, and our answers are in red. After addressing the comments raised, we believe that the manuscript would be sufficiently improved and reach the standards of TC.

**Comments from referees:**

**- Referee #2:**

How were the cutters sharpened for this study? Was the rake surface polished? I expect cutter sharpness and rake surface finish to impact cutting performance and cutting force.

This has been modified. In section of 2.2, we have explained the machining methods of cutters and the surface treatment methods of cutters after machining, and discussed the measure taken to avoid the impact of cutter roughness and grinding on experimental results.

I would like to see a wider range of rake angles investigated that include existing drills. Negative rake angles are not discussed in this paper but could provide an interesting comparison.

In the process of studying the influence of the bit geometry on cutting force, the angle of the cutter should include or even exceed the angle range already exists in the stage as much as possible. In this paper, we are committed to exploring the establishment of mechanical models during the ice cutting and drilling process, and verifying the correctness of the mechanical models within a certain range. And, the various angles of the cutters are interrelated, as the rake angle increase, the relief angle or wedge angle needs to be decreased. When the relief angle is fixed at 15°, as the rake angle increases, the wedge angle will decrease, while as the wedge

angle decreases, the strength of the cutter will decrease, as shown in Figure 1. Conducting experiments with different cutting depths may require apply a large drilling pressure on the cutter. Therefore, in order to prevent damage to the cutter during cutting and drilling process, we did not select a wider range of rake angles in this study. In the future, we will conduct more extensive research, and during the research process, the rake angle range of cutter will be wider and negative angles will be considered.

[Figure]

Figure 1. Schematic diagram of cutter structure

I would like to see a more detailed analysis of how the ice fracture mechanics change with depth of cut. High speed imagery is only provided for one set of parameters. At a minimum, I would like to see a comparison of imagery between different depths of cut.

This has been modified. In Figure 6, we added images of ice cutting process at different cutting depths and conducted a comparative analysis of the cutting and drilling processes under these two conditions.

The force diagrams in Figure 7. does not account for the presence of a cutter shoe which limits depth of cut. During cutting, I expect Fp to shrink to zero once the shoes are fully contacting the bottom of the borehole behind the cutter.

This has been modified. In Fig.7, we have added a schematic diagram of the relationship between the shoe and the cutting depth during the drilling process. In

the following text, we have provided a detailed explanation of the relationship between the shoe and $F_p$: "During the cutting and drilling process, the cutter comes into contact with the ice sample before the shoes. Only when the cutter is inserted into the ice layer with designed cutting depth, the shoes will fully contact the the bottom of the borehole. Prior to this, there will be continuous $F_p$ on the cutter. As the drill bit rotates, the cutter always inserts the ice sample before the shoe, and the $F_p$ on the cutter will continue to exist".

I would like to see a comparison of the cutting force trace (Figure 9) for different depths of cut. It would be interesting to compare the resulting frequency as chip size changes.

This has been modified. In Figure 9, we add a comparison of the cutting force generated at different cutting depths. And in the following text, a comparative analysis was conducted on the trace of the cutting force for different cutting depths.

On behalf of co-authors,

Rusheng Wang

---

## Author Comment (AC2)

Dear Referee,

Thank you for your response, and pass our great thanks to you for fruitful comments and advises. We tried to consider all mentioned comments and hereafter explain every change made point by point. The comments are in black, and our answers are in red. After addressing the comments raised, we believe that the manuscript would be sufficiently improved and reach the standards of TC.

**Comments from referees:**

**-Referee #1:**

The most interesting finding to me was seen in Figure 9 and discussed in lines 246 to 254; the low frequency oscillations from the formation and clearing of ice chips. I would like to see this concept expended – inherently all ice drills deflect under load so is it possible the oscillating cutting force creates a harmonic in the drill? Does this effect core quality? Can the oscillation be mitigated? Or could the drill be tuned to minimize cutting power by harnessing the momentum created with the ice fractures?

This has been modified. The 5.1 Analysis of the characteristics of cutting force section has conducted in-depth analysis and research on this oscillation. "Unlike ductile materials, where the chips produced by a shearing action are continuous and the forces encountered relatively constant, chips from brittle materials are produced by a repeated series of breaks, when the cutter presses into the ice, the force begin to rise and elastic energy is stored in the cutter assembly. Some of the energy is expended in local crushing as the force continues to rise. As the cutting force reaches a magnitude necessary to induce a major fracture. The ice layer undergoes shear-slip deformation. As the cutting force reached a magnitude necessary to induce a major fracture. A crack propagates into the ice, releasing the cutter elastic energy and dislodging a major ice chip, the force than suddenly decreases, before the cycle repeats. Therefore, during the cutting and drilling process in the ice layer, the cutting force exhibits an oscillating state, the frequency of the cutting force fluctuation should vary with the rotation speed of the drill bit.

And the amplitude of the oscillation is related to the cutting depth, as the cutting depth increases, the amplitude increases. At the same time, as the cutting depth increases, the degree of crack propagation into the ice will also increase. When the crack extends into the ice core, it will cause a decrease in the surface quality of the ice core. Therefore, it is necessary to control the cutting depth reasonably during the drilling process to ensure the quality of the ice core. Moreover, the study results on mechanical models of ice cutting process indicated that: "within the range of $\beta - \gamma_0 \leq \frac{\pi}{2}$, the cutting force gradually decreases with the increases of the rake angle". Therefore, the rake angle can be appropriately increases within this range to reduce the oscillation.".

I would like to see the effect of ice properties explored against the cutting force. We know that the ice properties change significantly with grain size and temperature (Petrovic, J.J., 2003) but this was not included in the study, although a robust drill design must work in a variety of conditions.

In the process of polar drilling, a robust drill design must work in a variety of conditions, and the drilling process under variety working conditions may require different cutting forces. And, we are committed to exploring the changes in cutting force under various experimental conditions. Therefore, an explanation of the reasons for selecting only one type of ice has been added to the article: "This study mainly focuses on the establishment of mechanical model during the ice cutting and drilling process. And, studies have shown that the crystal orientation, the crystal size and the density of ice samples in NGRIP boreholes in Greenland are similar to naturally formed and artificially frozen ice samples (Center for Ice and Climate, 2023; Cuffey and Paterson, 2010). Moreover, many scholars have conducted experiments on artificially prepared or naturally formed ice samples, and have ultimately obtained convincing experimental data and conclusions,

providing valuable references for research in the polar field (Narita et al., 1994; Talalay, 2003; Hong et al.,2015; Wang et al, 2024). In order to better observe the formation process of ice chip, at present stage, this study selected transparent ice and explored the fracture process and cutting force generated by this type of ice. And the ice with variety properties belongs to brittle materials, and there will be similarities in the fracture process. In the future, the cutting and drilling experiments under different ice sample properties to explores the effect of ice properties against the cutting force will be carried out."

The range of rake angle was from 20° to 40°, which doesn't even include the range of existing drills. Plots, such as Figure 10, indicated that the cutting force will continue to decrease with increasing rake angle. I would have liked to see that explored, at least until a physical limit was approached (e.g. the rake angle intersected with the relief angle). Then future designers could pursue better blade design and weight it against durability.

In the process of studying the influence of the bit geometry on cutting force, the angle of the cutter should include or even exceed the angle range already exists in the stage as much as possible. In this paper, we are committed to exploring the establishment of mechanical models during the ice cutting and drilling process, and verifying the correctness of the mechanical models within a certain range. In future research, we will expand the range of angle selection and even consider negative rake angles.

Same with depth of cut – it would be useful continue the trend to a maximum or asymptote.

It is very meaningful and necessary continue the trend to a maximum or asymptote, we are now in the gradual exploration stage, and in the future, we will conduct in-depth research on this, studying phenomena under various extreme conditions.

On behalf of co-authors,

Rusheng Wang

---

## Author Response (AR2)

Dear Referee,

Thank you for your response, and pass our great thanks to you for fruitful comments and advises. We tried to consider the comments and hereafter explain every change made point by point. The comments are in black, and our answers are in red. After addressing the comments raised, we believe that the manuscript would be sufficiently improved and reach the standards of TC.

**Comments :**

Thank you for your detailed responses to the reviewers' comments and updates to the manuscript. You added that sandpaper with gradually decreasing particle size was used to manually polish the cutter. Please include what the final particle size used was.

This has been modified. The particle size of sandpaper used for polishing cutters was explained in the paper.

Please consider adjusting your reference list with the next revision of your manuscript.

This has been modified. We have made corresponding adjustments to the format of the paper.

On behalf of co-authors,

Rusheng Wang